# An Institutional Perspective for Evaluating Digital Transformation in Higher Education: Insights from the Chilean Case

Karen Núñez Valdés [1,*] , Susana Quirós y Alpera [2] and Luis Manuel Cerdá Suárez [2]

1 Faculty of Education, Universidad de Las Américas, Av. Manuel Montt, 946, Sede Providencia, Santiago 7500975, Chile

2 School of Engineering, Faculty of Business and Communication, Universidad Internacional de La Rioja (UNIR), 26006 Logroño, Spain; susana.quiros@unir.net (S.Q.y.A.); luis.cerda@unir.net (L.M.C.S.)

* Correspondence: k.nunez.valdes@gmail.com

**Abstract:** From a managerial perspective, the rapid diffusion of actions and strategies accelerating the digital transformation of institutions is critical for success. However, in education, business, and management studies, digital transformation can be understood as simple evolutionary processes that enable business models, operational processes, and experiences to be made quickly and efficiently by institutions and agents. This aspect can sometimes lead to opposition, especially when little information is available or in situations of high uncertainty. This research aims to evaluate the involvement of an institutional ecosystem in the digital transformation at universities. Using data collected in Chile, this paper analyzes how the adoption of technologies by universities provides a context for understanding digitalization, measured by the IAU World Higher Education Database (UNESCO). The main finding of this paper is that there is a wide and relevant range of impacts of technological change in higher education institutions, particularly in the categories of values and operations. Additionally, this work serves as a repository of knowledge applicable to similar situations considering the specificities of each particular case. The importance to intervene in relation to certain variables at different levels of managerial performance is described and the implications for higher education institutions are discussed in these pages.

**Keywords:** digital transformation; higher education; Latin America; Chile; institutionalism

## 1. Introduction

Recently, the traditional characterization of information and communication technologies has focused on disruption and innovation [1] since digitalization has been observed (AI -artificial intelligence-, IoT –Internet of Things–, cloud computing solutions, among others) [2], including operational activities with a creative dimension, such as gamification, machine learning, virtual and augmented reality, and videogames [3,4]. Regarding this phenomenon, the pandemic and the different approaches taken by governments to enact lockdowns have accelerated the volatility, uncertainty, complexity, and ambiguity (VUCA) across the political, social, economic, and technological environment [5,6].

In this context, universities have been experiencing a set of relevant changes induced by the social and technological trends towards digitalization [7]. The adoption of technologies by universities is related to a paradigm shift [8,9] where technology is configured as a complex environment that enables digital learning, facilitates several learning experiences, and improves teaching materials and the training process in general [9–11]. For this reason, educational institutions have risen as relevant actors regarding innovation [12].

This paper refers to the digital transformation as a necessity in higher education institutions [12,13], and presents evidence of the relevance and impact that it has in the Latin American context. We identify the main disruptive technologies and explain how models and the processes of institutions are being transformed, and how they generate deep but somehow fragile changes. From this assertion we describe the challenges, risks,

opportunities, and guidelines that universities in Latin America must face to address their own digital transformation, particularly in the Chilean case. Digital transformation is considered to be inevitable [9,14,15], and it must be addressed with a clear understanding of each institution's peculiarities [16,17]. Additionally, in relation to the Chilean case, this paper outlines some ideas about the role that higher education institutions in Latin America could play to make contributions to the building of a more robust society, and what the strategy is that institutions are putting into practice to act as relevant facilitators of this change.

As claimed by several authors [9,18–20], digital transformation must be established according to a multidimensional perspective in order to meet the expectations of different interest groups in the economic, social, environmental, and cultural dimensions. For example, in an economic sense, the digital transformation generates a number of benefits, such as more profitable business models, more efficient operating processes, an enhanced value proposition, automated processes, and a lower cost of the exchange of information [21]. This consideration is becoming pertinent for universities, as competition to select the best students and researchers is increasing according to sustainable management.

Additionally, the social dimension that affects digital transformation is focused on the students experience and their lifecycle, and how digitalization improves both by replacing traditional educational models and services. What is more, this phenomenon is considered as a resource to create additional and differentiated value integrating digital technologies in teaching, learning, and institutional practices.

Moreover, the use and development of clean technologies also known as environmental, or green technologies [22], refers to the process that reduces negative environmental impacts through the sustainable use of resources and protection activities based on the evolution of information and communication technologies hosted online in the cloud, thus eliminating additional physical devices and hardware [23].

In the cultural dimension, learning can be categorized into three broad epistemological frameworks: objectivism, pragmatism, and interpretivism [23,24]. While objectivism points out that reality is external to the mind and understanding is acquired experientially, pragmatism states that knowledge is a tradeoff between action and inquiry. Nevertheless, interpretivism considers knowledge as an internal construction through socialization and cultural cues. Following a systemic perspective with regard to institutions, a fourth framework indicates that knowledge is composed of connections and networked entities [9,25], that is, an emergent, connected, and adaptive knowledge related to the epistemological framework for understanding the digital transformation in higher education.

However, considering that universities are organizations included in an ecosystem, nowadays it is even more important to understand the value that digital transformation initiatives may bring, not only to these higher education institutions, but also to other groups of interest shaping the education market. Certainly, assessing the maturity level that educational institutions have in their digital transformation processes may be of great relevance. The purpose of this paper is to offer an institutional perspective on the digital transformation in higher education [26] in order to understand the most relevant elements of the adoption of new technologies and their impacts in different educational institutions in the Latin American context, particularly in Chile. In the review of the literature that has been carried out, studies have been found that have addressed this issue [9,27]. For this reason, the aim of this study refers to the determination of whether the involvement of a particular educational ecosystem in the digital university transformation has had an increasing presence in the educational landscape by introducing innovative teaching methodologies linked to the digital transformation.

Given the exploratory and descriptive nature of this research, in this work it is hypothesized that digital transformation in higher education is built on a systemic approach that considers a range of voices for managing digital transformation in higher education institutions. Likewise, the issue of this study is to understand whether, among other variables, the social, organizational, and technological aspects are related to the interest generated by

the digital transformation in higher education institutions [27,28], or, conversely, to present the multiplicity of perspectives in which this digitalization has been addressed among different Latin American regions and nations.

While the digital transformation in the domain of higher education institutions is an emerging field that has aroused interest during recent years [9,28,29], little is known about how it operates in several regions and areas. Consequently, in order to conceptualize the systemic perspective for understanding this phenomenon, more information is necessary. Utilizing this approach, this paper explores how the adoption of technologies by universities provides a context for understanding digitalization, particularly in Chile.

The validity of this line of research is also determined by the fact that digital transformation has grown rapidly since the recent pandemic. Moreover, this study reveals that this phenomenon is based not only on scientific models, but also to a certain extent, chaotically, following several directions in this area. This is partially due to the lack of a scientific basis for building digitalization strategies in higher education institutions.

Based on references with a focus on regional ecosystems, our specific objectives are the following:

(1) to characterize the digital transformation in higher education as a set of dimensions providing a range of relevant voices regarding the role educational ecosystems play in this transformation;
(2) to evaluate the main dimensions for understanding this digital transformation from an institutional perspective in Latin America in empirical terms.

Assuming an institutional perspective in nature, a public, open consultation was carried out to investigate the current state of digital transformation in higher education particularly in Latin America, using data collected by the IAU World Higher Education Database (UNESCO). In this research, a mixed method in three stages was implemented as follows [9]: The first stage was based on conceptual work. Second, an instrument was carried out to measure the elements of the research proposal, which was applied to some higher education institutions, and finally, a general assessment of the results was performed when applying the proposal to universities, particularly in Chile. Additionally, the main insights of this study are focused on the conceptualization of the impact of this digitalization on innovation in teaching, learning methods, and an increasing agility reacting to changes and opportunities of a technology mix and its accelerated impact on Latin American societies, such as the Chilean case.

The conceptual and contextual framework regarding digital transformation in education is revealed in the next section. The theoretical approaches and details of the Latin American context, methods, and materials are described in the following sections, that is Sections 2.1–2.3 and 3. The findings related to digital transformation challenges in Latin American higher education institutions are highlighted in Section 4, regarding the Chilean case in particular. Summing it all up, conclusions and managerial and theoretical contributions are revealed in the last part of this paper.

## 2. Digital Transformation in Education: A Conceptual Framework and Context

Digitalization has gained relevance in the last decades as a consequence of the rapid evolution of technology and the telecommunication networks. This is due to the fact that digital transformation is a process that integrates digital technology in all aspects and requires changes in the areas of technology, culture, and institutions, among others; that is, technology is changing societies around the world in terms of the use of digital technologies, and this transformation has an impact on the skills and competencies required in society and the labor market.

The literature describing digital transformation has aroused great interest among researchers and practitioners, and many models have emerged which have tried to provide a framework to plan information and communication functions in organizations. One of the first models was the information systems architecture model [29,30], which provided a framework where data, processes, and business functions were considered in an integrated

way from different organizational levels (data flow diagrams and database modeling, in particular). From a management discipline, other models were incorporated to provide an alignment between information technologies and business strategy (the balanced scorecard method [31], for example).

On the one hand, the strategic alignment model emerged from the literature that included the relationships between information technologies, processes, and organizational strategies [9,32]. On the other hand, in another model the concept of maturity of the business of information technologies was introduced [33], allowing for the identification of pertinent steps for an organization to evolve in its alignment in terms of five levels of maturity based on objectives and governance issues, among others. Afterward, the term e-readiness was defined as the level to which a company is prepared to engage in electronic commerce [34], and it was later expanded to include the concept of digital maturity, defined as the capacity to have a potential evolution, implying the impact on businesses at the firm level.

There is a great variety of conceptualizations of digital transformation in the existing literature, from the application of digital maturity to business processes [35], that consider digital transformation to be disruptive [36]. However, many authors consider digital transformation to be the result of small but continuous digital innovations in terms of a function of accumulating digital innovations. One of the most relevant definitions describes it as an evolutionary process that takes advantage of digital capabilities and technology to enable business models, operational processes, and consumer experiences that generate value.

In agreement with the previous perspectives, some authors have attempted to identify the components of digital transformation and classify them as follows: drivers, objectives, success factors, and implications [37]. A framework drawing from an extensive review of the existing models was developed [9,38], including nine enabling factors that were classified into four categories related to organizational values, management capabilities, organizational infrastructure, and workforce capabilities, and this model was refined and applied to German organizations for validation. Similarly, the importance of providing guidance to managers on how to assess their level of advancement in digital transformation efforts was emphasized [39] through a six-dimensional model that included strategic vision, the culture of innovation, know-how and intellectual property, digital capabilities, strategic alignment, and technology assets. Empirically, their measurements were self-reported by managers, and they were based on measuring each dimension's progress in their particular company and comparing it with that of their competitors.

According to the literature, the rate of failure of digital transformation projects is 87.5% [40]. The authors identified the factors for success as familiarity with home office practices, availability, maturity of technology, and not needing to convince people that a change is necessary. Conversely, the causes of failure could be identified as unrealistic expectations, poor governance, and a limited scope. For this reason, a certain level of digital maturity in organizations and institutions is necessary. Additionally, digital transformation requires changes in culture and capabilities at different levels, depending on the degree to which the technology is used. Moreover, digital innovation in an organization is greatly dependent on two aspects: employee connectedness and responsive leadership [40].

## 2.1. Digital Transfomation in Universities

According to several authors, digital transformation in higher education does not merely refer to a technological transformation [41]. From an institutional perspective, the digital transformation in a broad sense is understood as a way to determine the stakeholder needs and behaviors in advance, and to provide education, research, and social services in line with the demands of the pupils who take advantage of the services in a changing environment. For this reason, digital transformation in education is being implemented worldwide step-by-step, with attention being paid to helping students with digital tools that can be reachable wherever there is an online computer terminal [41].

Saving time and resources by means of online management and tuition seems to be the consolidated challenge. This means the digitalization of core services, having academics and students with advanced digital capabilities, and decision support systems that can adapt to changing circumstances.

The present public health emergency due to a global pandemic has accelerated the pace during a mandatory lockdown. According to the above-mentioned literature and several authors in particular [6], three contextual considerations have arisen from this period. The first one points out that organizations must improve their digital maturity. The second one shows that less digitally mature organizations are more fragile. Finally, organizations are supposed to be generally more flexible with higher levels of digital maturity.

Bearing these aspects in mind, there is no doubt within this context that diverse social, organizational, and cultural backgrounds configure the transition to digital transformation in education, regarding not only contextual cues, but also several categories of social, organizational, and cultural situations. Historically, the role of a university has evolved from a former cultural role, through research-driven scientific advancement in the service of economic development to further optimize its own self-interest, to a brand-new social role ("the university for others") [42,43]. In the words of one author, "automation will make many jobs obsolete before long", therefore higher education institutions must meet the pace of digital transformation to survive and furthermore, deliver their subjects in a more flexible way, reinforcing their institutional function as organizations for change. Information is key for social development [44,45], and higher education institutions play the role of conveying it to society by using effective tools and strategies [9,46] after thorough managerial work before this role is performed by teachers. The information digital transformation challenge, according to universities, requires an adaptation to society's needs.

To promote educational digital transformation in terms of connectedness, students and professors linked through the internet or remote maintained machinery give a systemic and institutional perspective that is not new to educational institutions [46]. Although the decomposition of degrees into smaller open-source learning networks will provide the skills needed for a job, and every single second data emerges digitally from every action commanded on online sources, this massive amount of data must be understood within a context which makes sense of it [47,48]. To meet both academic and institutional needs, numerous groups all over the world are following the trend of using Artificial Intelligence that emulates human behavior [49] to forecast what actions will be needed according to data and environment dichotomy, and to give a response to every remote stimulus, which is the core of the Fourth Industrial Revolution in education [49,50]. In this line of research, digital transformation in higher education institutions comes from managerial work, supported by institutional structures based on human, organizational, and technological resources.

## 2.2. An Institutional Perspective of Digital Transfomation in Universities

Universities were the center for knowledge production and dissemination for centuries. These elements have been challenged over recent decades by a parallel ecosystem which plays the same role, based on the Internet [51]. The access to knowledge worldwide is no longer restricted to the physical space of the university, but it is found in different platforms, applications, encyclopedias, and open-source web browsers that allow people to learn about diverse issues, which is the trademark of the digital era. This new scenario represents a challenge rather than a threat for higher education institutions, especially in Latin America and the Caribbean (LAC), where transformations occur in a slower way in comparison with developed countries [52].

Besides the manufacturing industry, academic institutions are certainly involved in Industry 4.0 [53]. Although ancient wisdom is sheltered in digital libraries which collect the roots on which development is built, innovation means gathering different branches of knowledge and obtaining something new from them [54]. According to an institutional perspective, among all academic institutions, higher education institutions are prone to

give specific tuition in every discipline, exploring every field, making connections, and bridging the gaps between them.

However, several authors point out that academic institutions are often considered cutting edge centers, whereas diverse evidence shows that universities encourage mainly conservative and gradual research instead of audacious and innovating research [9,55]. According to this approach, business excellence in a volatile, uncertain, complex, and ambiguous environment (BEVUCA) [5] has much to contribute to higher education to fill the gap by considering the overall VUCA influence and the influences of each specific term individually [56]. New epistemologies and paradigm shifts have been proposed [57], according to Big Data [58–60], claiming "the end of theory" by promoting the creation of data-driven science, instead of knowledge-driven science, and developing the digital humanities, as well as the computational social sciences, which show alternative ways of approaching culture, history, economy, and society [61].

Because digital innovation in education is dependent on responsive leadership, considering higher education institutions as businesses that set relationships between stakeholders, mentors, and supports, is a recent model that has focused on developing managerial competences besides technical competences based on a comparison of technology maturity models [10]. This model is organized in six steps: Identification, Definition, Design, Development, Evaluation and Communication, offering an important guideline in this way:

1. Identification, where the competences required should span from the ability to monitor, analyze, and comprehend the benefits of technological trends, to the deep knowledge of the organizational business structures, processes, strategies, in order to disclose possible convergences between the two;
2. Definition, which includes the setting of the necessary resources and tools for starting the digital transformation process. The competences required should be at the organizational level (resource management, and so on), and at motivational level, so that a common vision of the process of transformation can be effectively communicated and shared among diverse actors;
3. Design, where the technical competences for integration purposes, as well as business process design competences for process re-design, are both necessary;
4. Development: Project management competences are highly required in this phase for the organizational and coordination aspects of the project;
5. Evaluation: in this phase data analysis capabilities facilitate the evaluation of the risks and the impact of the project in terms of a high volume of data that should be gathered, elaborated, interpreted, and communicated;
6. Communication. In this phase, a set of competences are related to leadership, communication skills, persuasion techniques, and the ability to gain approval for the project results.

*2.3. Understanding Digital Transformation in Higher Education: The Latin American Context*

The ability of universities to attain their objectives following these steps is usually restricted by contextual constraints in the political, social, and economic dimensions. Authors claim that the strategic challenges for the universities in the Latin American region in particular are, first, taking advantage of the synergies among the members of the university community with the integration and articulation of their own special areas of university work (teaching, research, outreach, and engagement activities) through the use of technology and the abandonment of monopolistic logic in order to question themselves frequently and compare themselves with their environment [9,59] to determine how their students can learn more and learn better, and what technologies could help the development of their students [60–62].

Adopting an institutional sense, the Inter-American University Organization (IUO) proposes integration and cooperation among higher education institutions from Latin America and the Caribbean to develop joint working agendas based on the challenges that appear, which must be taken up by these institutions. In this way, the IUO intends to act as

a facilitator of the changes that must be made in higher education. Research conducted by different authors reveals three lessons as introduced above, that the period of the pandemic has particularly highlighted regarding digital transformation [6]: (1) organizations must improve their digital maturity; (2) institutions that are less digitally mature are more fragile; and (3) organizations or corporations with higher levels of digital maturity are generally more flexible. These lessons can provide guidelines for higher education institutions on how to face this context, which includes digitalization, artificial intelligence, and industry 4.0, for their functioning [63].

The diagnosis made in this research shows the urgency of developing strategies that accelerate digital transformation and the full inclusion of the region in terms of technologies, especially because it is expected that more than 20% of the jobs from some countries, which are part of LAC, will undergo some kind of automation, which demonstrates the need for new investments in education and training to equip workers with the necessary digital skills to cope with this new scenario [63–65].

To sum up, digital transformation must be understood as a process which has been accelerated by the health crisis caused by COVID-19, and that challenges universities to evolve towards models of organization based on continuous innovation, where it is necessary to redefine both the services oriented at students in the academic field and companies and organizations in the area of transfer [66].

A recent report has examined several aspects within the higher education institutions in order to assess the level of digital transformation within various facets [67], looking at changes from the perspective of the overall institutional governance, the use of technology in teaching and learning, reviewing the progress made towards the use of Open Educational Resources (OER), Open Science, and the availability of digital knowledge infrastructures such as an online library [68]. This report represents the first stock-taking exercise in the field, contributing to the discussion on the current state of the digital transformation in higher education from an institutional perspective. In this section, a focus on the two first facets are described.

## 3. Materials and Methods

There are several review methods for analyzing and evaluating the existing literature, such as a critical review, a literature review, a meta-analysis, a systematic search, and a review [61]. The present work was performed in three stages: the first one was based on conceptual work. Second, an instrument was created and validated to measure the elements of the research proposal, which was applied to some higher education institutions, and finally, a general assessment of the results was performed when applying the proposal to universities.

In empirical terms, the research presented in this paper is part of the larger research project regarding the results of the Open Consultation carried out by the International Association of Universities (IAU) from 1 November 2018 to 1 April 2019 [62]. A consultation was carried out to take stock of the current state of digital transformation in higher education around the world, and to inform the development of a new IAU Policy Statement. Firstly, this report focuses on the national context in which higher education institutions are operating to assess to what extent higher education institutions are operating in an environment conducive to digital transformation. Moreover, this study examines both the infrastructure and governance in terms of policies and educational regulations. Secondly, this is followed by a section looking at changes as they relate to teaching and learning.

### 3.1. Population and Sample

In this research, an Expert Advisory Group was established composed by IAU Board members and experts from different parts of the world, bringing together a broad range of expertise both in higher education leadership as well as in specific areas of digital transformation. With the aim to characterize the digital transformation in higher education as a set of dimensions, and to evaluate the main dimensions for understanding this digital trans-

formation from an institutional perspective in Latin America, the data collection procedure was divided into two separate consultations: (1) the targeting of the leadership of higher education institutions (i.e., leadership consultation), and (2) open to all representatives in higher education institutions across the institution (that is, comprehensive consultation). This procedure was based on the desire to reach out to the leadership of these institutions. Additionally, it was also decided that several representatives from the same institution could facilitate information from as many different sectors as possible within institutions that may be involved in activities or initiatives related to digital transformation.

The IAU World Higher Education Database (WHED Portal, www.whed.net, accessed on 23 November 2020) constituted the more relevant source of information to reach out to higher education institutions worldwide. In collaboration with UNESCO, this IAU WHED Portal provides information on higher education systems in 196 countries and territories, and over 20,000 higher education institutions. For the launch of the consultation in 2018, heads of institutions and representatives received an invitation to participate in the consultation through their national networks and rectors conferences, including UNESCO, the Commonwealth of Learning (COL), the European University Association (EUA), the Association of Universities of Latin America and the Caribbean (UDUAL), the Association of African Universities (AAU), the Groningen Declaration Network (GDN), and the Conference of Rectors of Universities (CRUE), Spain.

### 3.2. Measurements

Certainly, the digital transformation is linked to the fourth industrial revolution and involves the adoption of new skills of individuals. On the one hand, these new learning spaces include Artificial Intelligence (AI) as a relevant educational resource, adding value and allowing students to discover new teaching methods. However, on the other hand, institutions are using AI in order to personalize the student admission process and help the teacher identify student progress.

Consulting stakeholders is an important instrument to collect information for social problems and evidence-based policymaking. Their views, perceptions, and experiences help deliver higher quality and more credible policy initiatives and solutions, evaluations, and checks. It also ensures greater transparency and the legitimacy of the policy development process and contributes to a more successful strategy implementation.

In this research, the purpose of the open consultation was to design an efficient and effective consultation approach. The consultation strategy aimed to ensure that all relevant evidence was considered, including data about antecedents, impacts, and the potential benefits of the specific initiative. In this context, validation means a process of confirmation by an authorized body that gave results, and the findings were measured and consisted of the following four distinct phases: (1) identification through a dialogue of the experiences of stakeholders; (2) documentation to make visible the individual's experiences; (3) an assessment of these experiences; and (4) certification of the results which may lead to a partial or full qualification.

Following the aim to collect diverse data according to the requirements of this study, the participants in this research were selected from a range of higher education institutions as evidence of triangulated information to improve the validity of the research by collecting geographically distributed data. In order to have important results and conclusions, this data collection considered opinions that represented the main changes of educational practices, rather than being a consequence of the specific context. The reasons for this kind of design among the institutions and contexts were related to the fact that relevant aspects in corporative cultures and educational practices provide insights on each specific situation. In this sense, differences and similarities can be discerned and this methodology can provide information on a theoretical basis as well as implications for practice, particularly for each situation.

Working jointly with private and public sectors, civil society, and academia, the Inter-American Development Bank designed an initiative to promote the responsible adoption of

Artificial Intelligence (that is, fAIr LAC) in order to improve the delivery of social services and create development opportunities [63]. One of the first steps of this initiative was to document a greater amount of information on the progress made in the field of AI in higher education to provide evidence of the digital transformation in universities, and on relevant use cases in Latin America and the Caribbean (LAC).

Based on a subregional assessment and several criteria including digital maturity, international rankings, and the progress achieved in AI, among others, this initiative included a diagnosis of the current situation of AI in Argentina, Brazil, Chile, Colombia, Costa Rica, the Dominican Republic, Ecuador, Mexico, Paraguay, Peru, Trinidad and Tobago, and Uruguay.

*3.3. Methodological Process of the Study*

Focused on documentary analysis and information shared by local and regional experts for each country, the information collected is organized in the next pages under the following subsections [9,63,64]: (1) a summary of the country profile as a general context; (2) documentation of the different efforts made by government, academia, the entrepreneurial sector, and civil society to strengthen and develop an AI ecosystem in the region; and (3) a general conclusion. Given the exploratory character of this research, relatively free form answers from respondents were collected (data collection procedure) in a less structured manner with the following information regarding each particular ecosystem:

Part I: institutional and contextual elements linked to digital transformation (that is, the external environment of institutions);
Part II: digital transformation in higher education at the institutional level;
Part III: perceptions of the developments related to digital transformation in higher education, and society in general.

In this scenario, LAC has had to face the crisis caused by the coronavirus (COVID-19), which entailed the impossibility of physical contact due to the quarantine period that was implemented in different countries [65]. This new reality made digital technologies a priority, especially because these enabled part of the population and companies to continue studying and working while sanitary measures were complied with. For international organizations, the global pandemic "made inclusive digital transformation a top priority to mitigate the negative effects and accelerate economic recovery of countries" [66]. Likewise, the pandemic revealed the difficulties that organizations with low levels of digital maturity have had to face; organizations have had to work in a context of adversity, uncertainty, and of great fragility [6], testing the capacity of global economies to adapt and, particularly, of higher education institutions in the region.

The study focused on the graphic visualization and numerical analysis of the relationship between categorical variables (questions in Part I and Part II). Additionally, several scales were defined with a Likert-type scale ranging from 1 strongly disagree to 10 strongly agree (questions in Part III). At the same time, we connected the perceptions of the participants in this research with the geographical region of their institutions for comparison. Categories were defined and frequencies were obtained from the results of the proposition hypothesized. Regarding the perceptions of the developments related to digital transformation, (Part I, Part II, and Part III), in order to analyze the relationships between variables in operational terms, the following null hypothesis was tested in all cases: $H_0$: There is no association between the institution's perceptions and the geographical regions being considered. Hypothesis $H_0$ was tested by making statistical contrasts between the associational relationships of the all variables. To that end, we used Pearson's chi-squared test. To determine the strength of a relationship after a chi-squared test revealed a statistically significant association, we also performed a contingency coefficient and Cramer's V test. In this research, the Mann–Whitney and the Kruskal–Wallis H tests were used to compare populations in order to contrast whether the perceptions of institutions surveyed on digital transformation differed significantly.

## 4. Results and Discussion

Although several analyses in this research were quantitative in nature, the main aim of this section is to provide a general framework to aid in the qualitative interpretation of the results. Thus, it is important at this moment to describe the empirical insights for further exploration and discussion.

According to the methodological process described in this study, Table 1 shows the profile and distribution of replies worldwide (descriptive statistics). IAU received a total of 1039 complete replies from 127 countries (replies by region are listed below).

**Table 1.** Profile and distribution of replies by region.

|  | **Africa** | **Asia and the Pacific** | **Europe** | **LAC** | **Middle East** | **North America** | **Total** |
|---|---|---|---|---|---|---|---|
| Leadership consultation | 70 | 75 | 119 | 43 | 27 | 13 | 347 |
| Comprehensive consultation | 146 | 186 | 202 | 38 | 117 | 3 | 692 |
| Total | 216 | 261 | 321 | 81 | 144 | 16 | 1039 |

Source: [62] and own elaboration.

### 4.1. Analysis of Results and Discussion: Implications for Latin America

Regarding the above-mentioned methodology of the study, the findings described in these paragraphs derive from data analysis. First, we describe the different elements to understand the conditions for digital transformation at higher education institutions and universities in terms of the context and the factors of relevance in the governance of institutions.

Respondents were asked about their general perceptions regarding the relevance of national higher education regulations and about their opinions concerning the digital transformation. Figure 1 shows the bar charts of participants' opinions to these contextual elements for four answer categories (from a = highly supportive, to d = mostly unsupportive). In terms of Part I, in the leadership consultation, regarding the question: 'Are national regulations conducive to digital transformation?', the regions weighing higher on the negative side of the scale were Europe (59%), Africa (55%), and LAC (53%) where most respondents considered the national regulatory policies less conducive to digital transformation in general (that is, adding 'variably supportive' and 'mostly unsupportive; Figure 1).

Respondents were asked about their general perceptions regarding the relevance of the national internet infrastructure (Figure 2 shows the bar charts of participants' opinions to these aspects for six answer categories, from a=very satisfactory, to f=other). In terms of the comprehensive consultation, respondents were questioned with a focus on internet infrastructure and the situation for higher education institutions. In this sense, there was a relevant difference between Europe (39% described the internet infrastructure as satisfactory), and LAC (58% said 'good in big cities, but poor in rural areas'). Certainly, this consideration suggests that the opportunities are not the same in terms of digital transformation (Figure 2).

Additionally, in the comprehensive consultation, respondents were also asked to describe the internet infrastructure within the institution. Figure 3 shows the bar charts of participants' opinions to this question for four answer categories (from a=yes, a significant obstacle, to d=other). The results illustrate the different conditions of institutions: the respondents in Europe indicated that infrastructure was not an obstacle (68%) in contrast to LAC (58%: 'Yes, a significant/minor obstacle'; Figure 3).

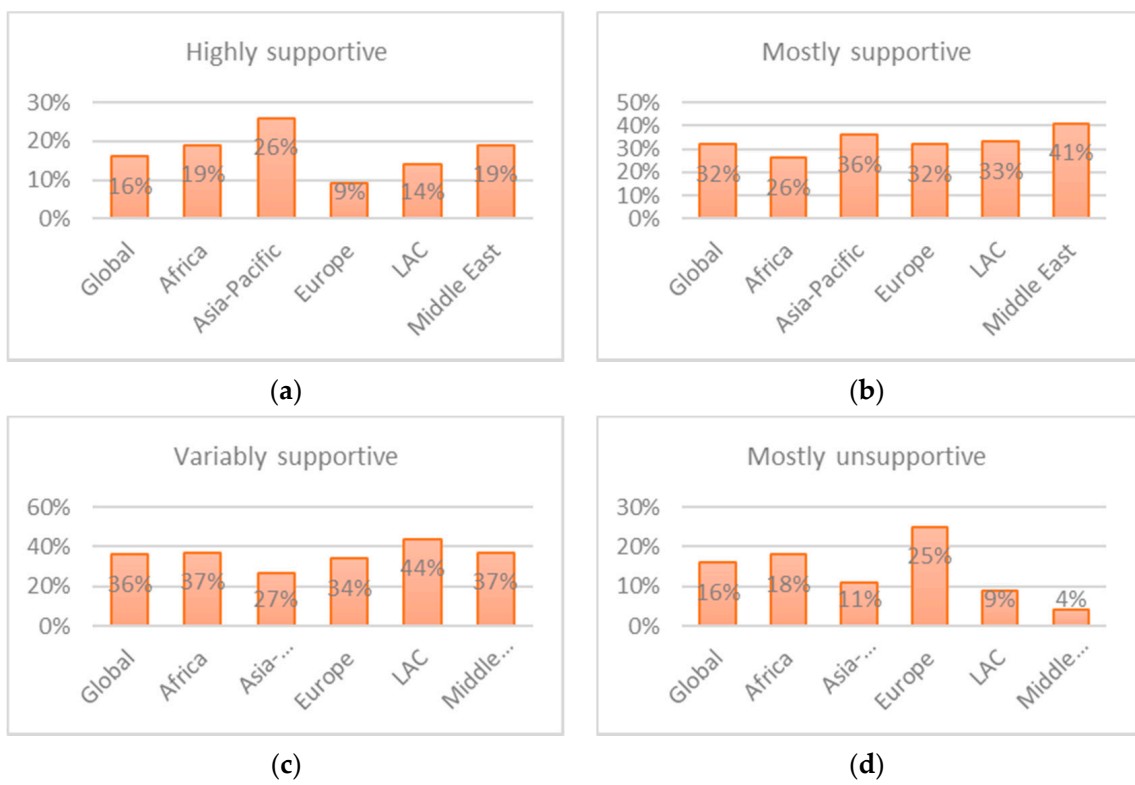

**Figure 1.** Relevance of national higher education regulations. Source: [62] and own elaboration.

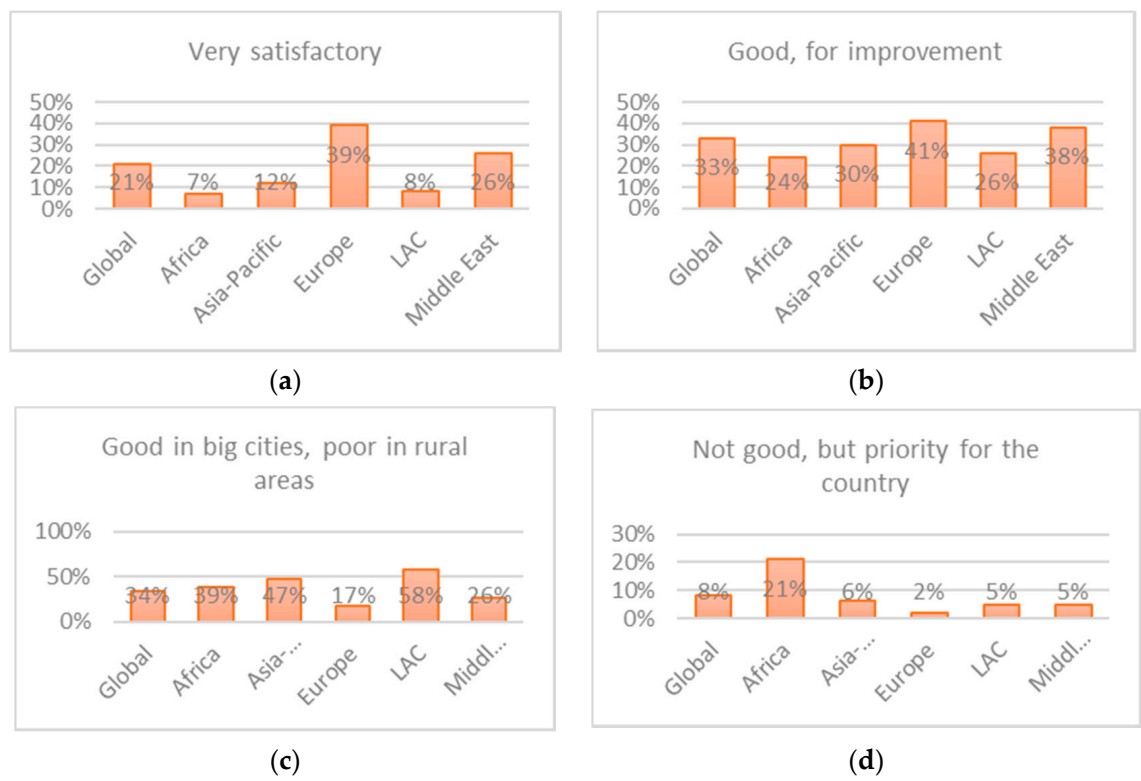

**Figure 2.** *Cont.*

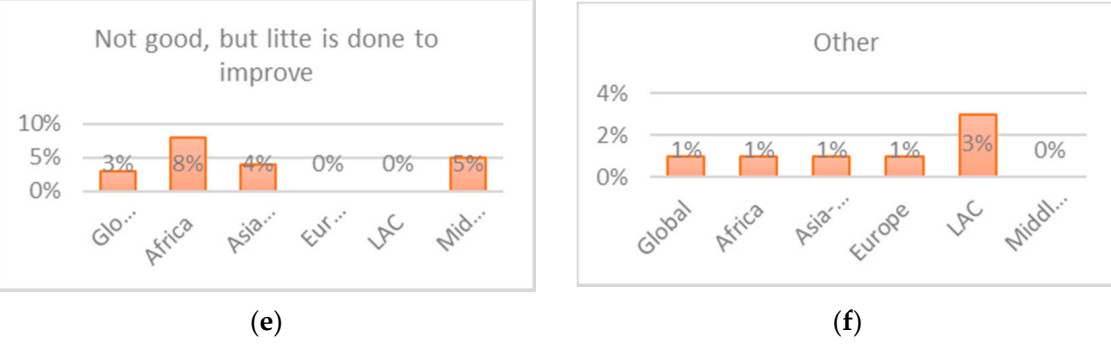

**Figure 2.** Relevance of the national internet infrastructure. Source: [62] and own elaboration.

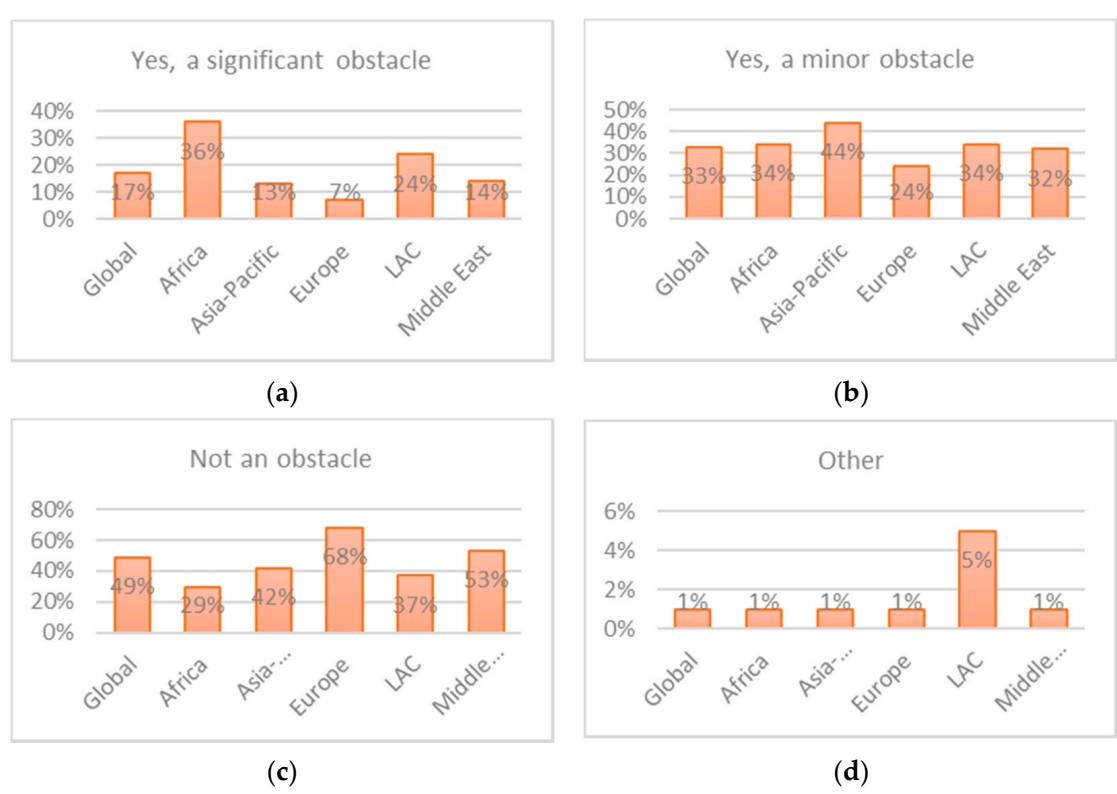

**Figure 3.** Relevance of the digital infrastructure at institutions. Source: [62] and own elaboration.

Respondents were asked about their general perceptions regarding digital transformation in higher education. Figure 4 shows the bar charts of participants' opinions to this consideration for four answer categories (from a=an institutional strategy with an institutional vision, to d=other). In terms of Part II, regarding the digital transformation in higher education at the institutional level, in the leadership consultation in relation to the query: 'How is digital transformation translated into action at your university?', two answer categories stand out. The first one referred to an institutional or national strategy with a clear vision for the institution, led by LAC and Europe (49%), and followed by Asia Pacific (39%), Africa (34%), and finally the Middle East (30%), as shown in Figure 4. The second one was related to the direct action of faculties and departments pursuing multiple initiatives according to needs and opportunities, with the Middle East first (70%) followed by Africa (63%) and Asia Pacific (55%), whereas LAC and Europe remain tied (49%), showing the lowest result for this answer.

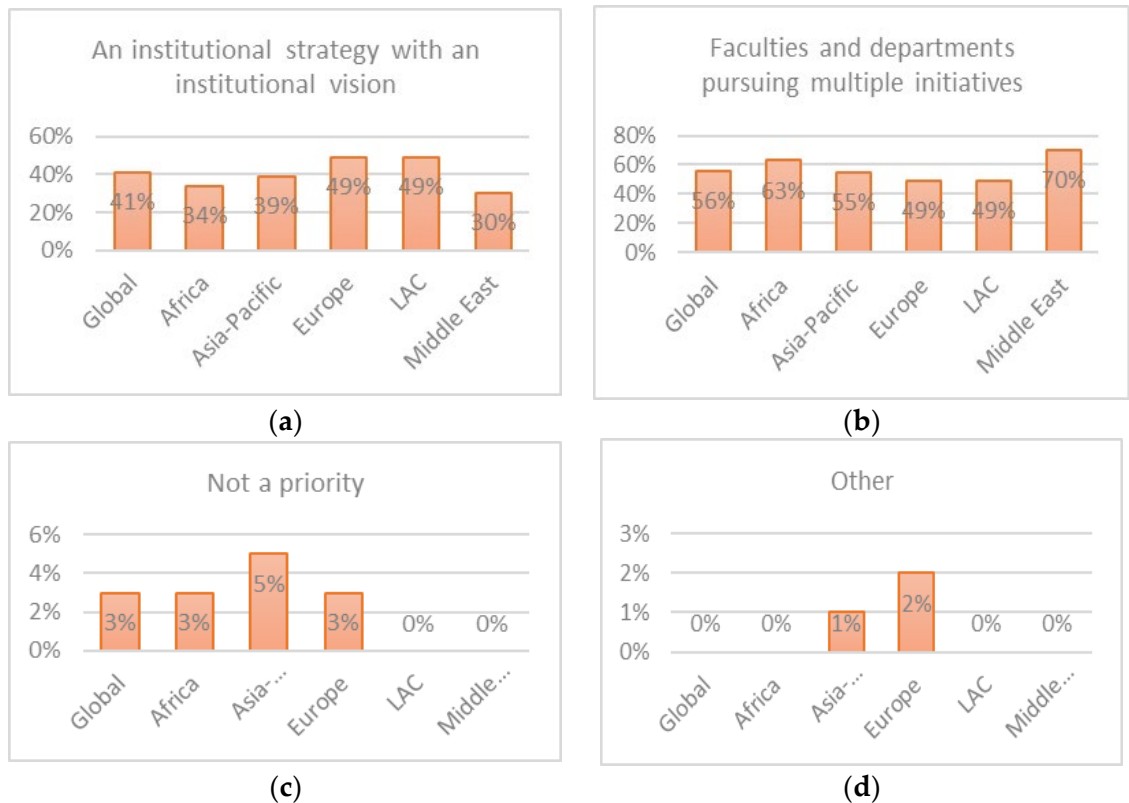

**Figure 4.** Digital transformation translated into action at your university. Source: [62] and own elaboration.

In the comprehensive consultation regarding how digital transformation translates into action through a different set of measures, the respondents were asked whether the use of new technologies is part of the institutional strategic plan (Figure 5 shows the bar charts of participants' opinions for six response categories (Figure 2 shows the bar charts of participants' opinions to these aspects for four answer categories, from a=yes, to d=not applicable). In this category, Europe (13%) and the Middle East (12%) were above the global average. Globally, 13% have stated that they 'don't know', and in this category the LAC were above the global average (18%).

Respondents were asked about their general perceptions regarding the relevance of the technology integrated into teaching. Figure 6 shows the bar charts of participants' opinions to these aspects for four answer categories, from a = yes, fully engaged, to d=not at this stage). In the comprehensive consultation, regarding technology integrated as part of teaching, the main responses were 'Fully engaged' and 'To some extent'. In Figure 6 it is clear that the Middle East took the lead (40%), but LAC showed the least percentage (11%) in fully integrating technology into teaching. Nevertheless, as shown in Figure 6, LAC led in relation to the integration of technology to some extent (79%), with Africa in the last position (49%).

Particularly in the LAC, international organizations pointed out that the digital transformation needs to be thought about in a systemic way and in the long term, with a consideration of the historical transition of this region and its peculiarities, where changes are developed in a slower way compared to other regions globally [67]. While there has been a substantial transformation in recent years, there are still digital divergences in both homes and companies, which leaves the most vulnerable segments of each country behind [66]. Similarly, digital transformation has been developed unequally, which is evident from: (i) 68% of the total population of LAC had access to the Internet in 2018, far below the average of the OECD countries, which reached 84% in terms of access in the same year, (ii) 75% of the richest population of Latin America uses the Internet and only 37% of the poorest population uses it; therefore, the difference between rich and poor is

much bigger (almost 40 percentage points) in LAC than in the OECD countries (fewer than 25 percentage points) [67]. This information reveals the difficulties that LAC has in moving towards the digitalization and inclusion of artificial intelligence in the region.

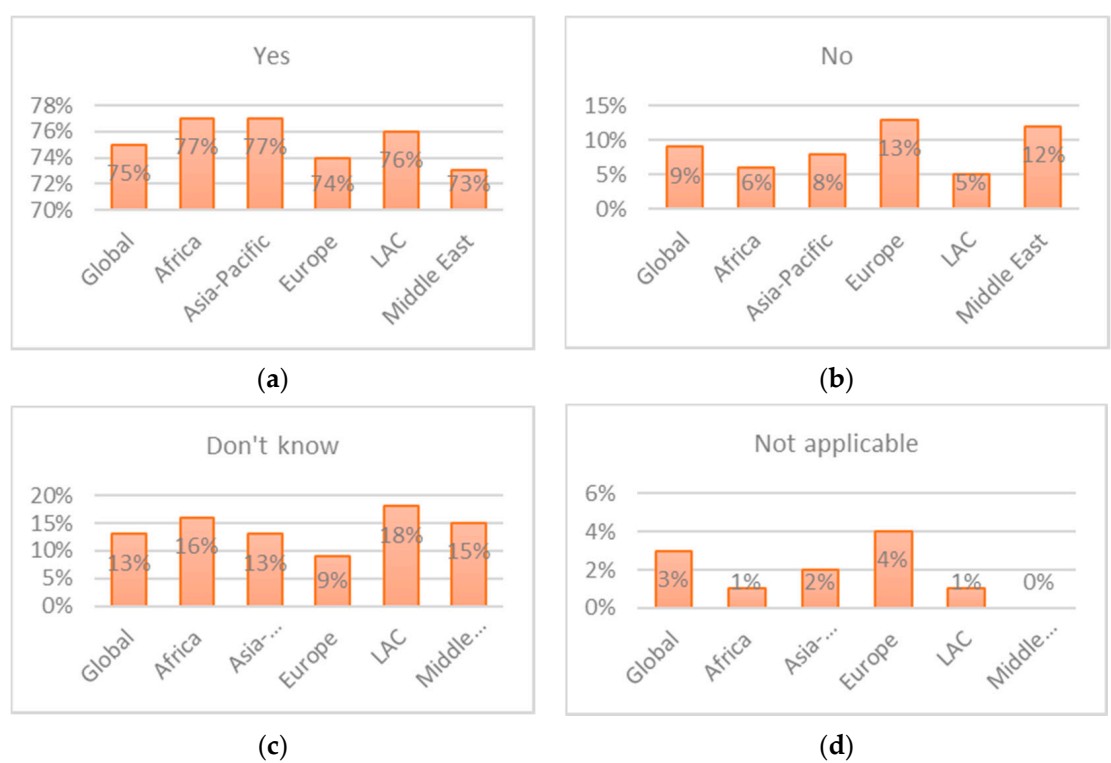

**Figure 5.** Digital transformation in the institutional strategic plan. Source: [62] and own elaboration.

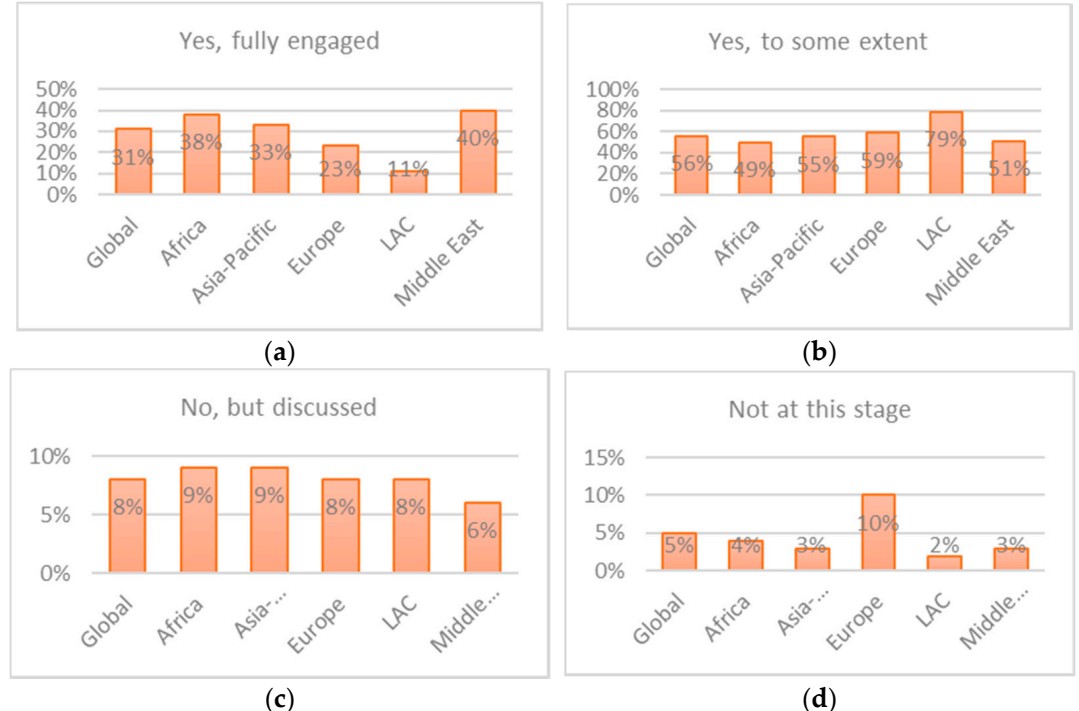

**Figure 6.** Technology integrated into teaching. Source: [62] and own elaboration.

Regarding the changes from the perspective of the overall institutional governance and the commitment to and the nature of change, this research aimed to assess to what extent leaders in higher education consider digital transformation as a priority; most respondents considered it a 'high priority (68%) or 'medium priority' (29%) and only very few considered it a 'low priority' (3%) or 'not a priority' (1%). Differently, in the comprehensive consultation, respondents were asked about the commitment of leadership towards digital transformation and the use of new technologies within the institution. In comparative terms, the results confirm that 72% of respondents found that there was a commitment from the leadership at the global level, particularly in Africa (77%), but less so in Latin America and Caribbean (61%) which were below the global average. Despite the differences, there was an overall trend where leadership considered 'digital transformation' an important priority and respondents confirmed this strong leadership commitment to pursuing digital transformation in most institutions.

Regarding the use of technology in teaching and the use of new teaching modalities, from a general perspective, respondents were assessed in relation whether technology was being increasingly integrated as part of teaching. Most respondents expressed 'yes, to some extent' (56%), 31% indicated 'yes, very much', 8% replied 'No, but it is being discussed', and only a few selected 'No, not at this stage (3%). However, there were some regional differences between the categories ('Yes, very much' and 'Yes, to some extent'). For Latin America, only 11% of the respondents answered 'yes, very much' in contrast to 79% who responded 'yes, to some extent'. Europe was also below the global average in the category 'Yes, very much' (23%) but above average in the category 'Yes, to some extent' (59%).

In terms of Part III, regarding the perceptions of the developments related to digital transformation in higher education and society in general, while digital transformation has been constant, the pace at which it is unfolding today is unprecedented. This process not only creates hope for the future, but also insecurities. For this reason, the author of [62] consulted university academics about their opinions related to higher education and technology. The leadership consultation began with the assumption that the integration of technology in higher education implies change and transformation; therefore, how prepared are the institutions to change and innovate? In this regard, 35% of respondents from LAC indicated that their institutions were very prepared and only 2% considered that their institutions were reluctant to change. This region was above Europe, Asia, and the Middle East, and below Africa (Figure 7).

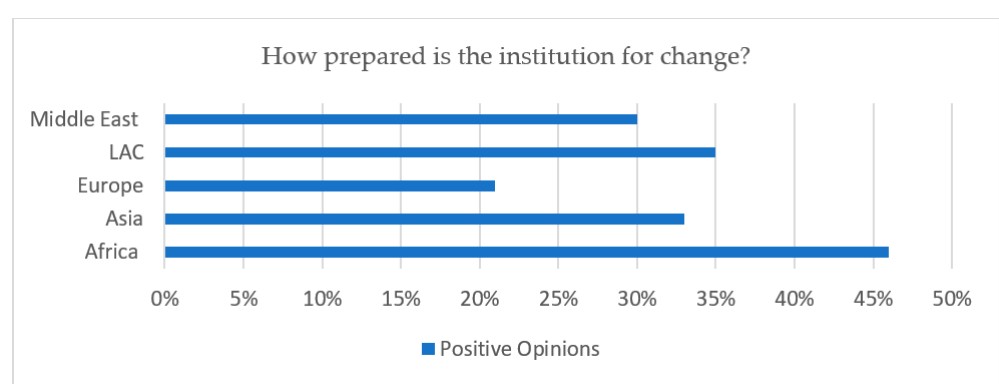

**Figure 7.** Preparation for change at your institution. Source: [62] and own elaboration.

The opinion of those surveyed about digital transformation and the need for students to actively participate in society is interesting. A total of 58% of respondents from the LAC region fully agreed that the preparation of students is crucial, and their responses were below the study average (61%). However, there was a consensus among the regions on the importance of this preparation in terms of the comprehensive consultation (Figure 8).

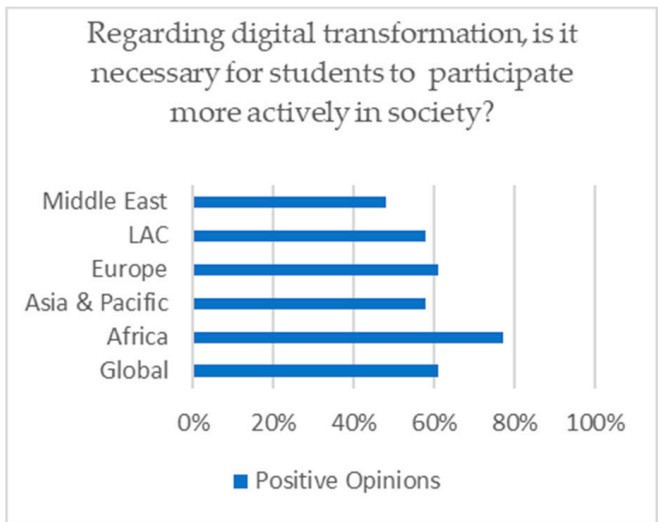

**Figure 8.** Relevance for students to participate in society. Source: [62] and own elaboration.

Regarding the potential of digital transformation to exacerbate socioeconomic gaps within and between countries, respondents in the comprehensive consultation considered that there was a significant risk associated with new technologies (Figure 9).

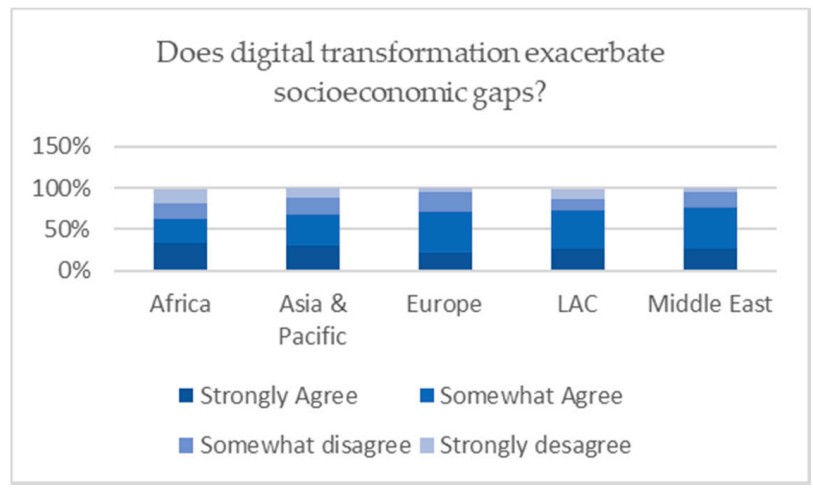

**Figure 9.** Potential of digital transformation to exacerbate socioeconomic gaps. Source: [62] and own elaboration.

Despite these opinions, there was a consensus among those surveyed about the importance of digital transformation and new technologies in improving higher education. The regions of Africa (90%) and LAC (86%) were the most optimistic about the future of higher education.

In addition to this graphic visualization, in order to further investigate our results, the distribution pattern of the data was observed by examining the results of descriptive statistics (that is, the responses to the questions were analyzed). The selection of the appropriate statistical test depends on the characteristics of the variables: chi-squared, contingency coefficient, and Cramer's V in this study. Mann–Whitney and Kruskal–Wallis tests are usually performed if the scale data are non-parametric, or if the nature of data is uncertain. The results of this analysis are presented in Table 2 for conventional levels of significance (*p*-Value < 0.05, where $H_0$ is rejected).

**Table 2.** Chi-squared statistics for global results and particularities by regions (Europe *versus* LAC).

| Questions | Chi-Squared (Sig., 2-Sided, *p*-Value) | Conting. Coef. (Value) | Cramer's V (Value) |
|---|---|---|---|
| Part I | | | |
| National regulations: global results | 0.000 | 0.262 | 0.301 |
| Europe vs. LAC | 0.001 | 0.300 | 0.311 |
| National internet infrastructure: global results | 0.059 | 0.260 | 0.288 |
| Europe vs. LAC | 0.000 | 0.345 | 0.401 |
| Digital infrastructure at institutions: global results | 0.003 | 0.206 | 0.259 |
| Europe vs. LAC | 0.001 | 0.244 | 0.402 |
| Part II | | | |
| Digital transformation into action: global results | 0.000 | 0.368 | 0.403 |
| Europe vs. LAC | 0.003 | 0.378 | 0.411 |
| Digital transformation in the plan: global results | 0.052 | 0.333 | 0.467 |
| Europe vs. LAC | 0.001 | 0.331 | 0.498 |
| Technology into teaching: global results | 0.000 | 0.277 | 0.288 |
| Europe vs. LAC | 0.000 | 0.275 | 0.323 |
| Part III | Mann-Whitney (*p*-value) | Kruskal-Wallis (*p*-value) | |
| Preparation for change–institution: global results | - | 0.051 | |
| Europe vs. LAC | 0.000 | | |
| Participate in society–students: global results | - | 0.031 | |
| Europe vs. LAC | 0.001 | - | |
| Exacerbating socioeconomic gaps: global results | - | 0.000 | |
| Europe vs. LAC | 0.000 | - | |

Source: own elaboration.

The results shown in Table 2 reveal that in global terms and for conventional levels of significance, the hypothesis of homogeneity in the perceptions among regions can be rejected and, therefore, we accept that institutions differ about the relevance of digital transformation in higher education depending on the geographical region surveyed. Additionally, toward the end of examining these perceptions more closely, we undertook a segmentation analysis of the sample between Europe and LAC in comparative terms. The results of the Europe versus LAC analysis allow us to conclude that these opinions are generally shared by the sample as a whole.

As a result, it is very clear that technology is increasingly being used in teaching. According to the author of [62], this can be a sign of a potential to integrate it more fully, or maybe that it is important that technology is only used somewhat in teaching. Moreover, in the leadership consultation, the respondents assessed to what extent they use teaching modes such as a flipped classroom, blended learning, and online learning. Twenty-seven percent stated 'Yes, fully' and 52% indicated 'Yes, somewhat'. Certainly, a general tendency towards more integration of technology through new teaching and learning modes was evidenced. In terms of the regional breakdown, Latin America had the highest score in 'yes, fully' (49%) and was below the average in 'yes, somewhat' (37%). Europe was slightly below the average in 'yes fully' (24%). However, when considering the sum of the 'yes' categories, both LAC (86%) and Europe (84%) were at the same level. Finally, the Middle East (22%/7%) and particularly Africa (29%/9%) were the two regions above the global average (16%/5%) in both of the 'no' categories.

Measuring the change in teaching pedagogies and approaches, the research assessed whether lectures continue to be the dominant form of teaching in higher education. One

possibility of technology is to disseminate information from one to many: for example, 'problem-based-learning' implies a more active engagement on the part of the students in the learning process. The regional breakdown showed that Asia, the Pacific (26%), and Latin America (24%) were above average in the category 'Mostly problem-based learning but combined with lectures' while the Middle East (32%), and Africa (27%) were above average in the category 'Lecture-based learning'. Africa (56%) and Europe (53%) were above average in the category.

Within this context, respondents to the leadership consultation were asked to assess whether the university has reconsidered the skills and competencies required of students within the past 3–4 years. A total of 82% of respondents indicated 'yes', where 35% of replies were 'yes, fully' and 47% were 'yes, somewhat'. Thirteen percent responded 'no, but it is being discussed' and 5% said 'no, not at this stage'. Replies from Asia and the Pacific and Europe followed the global trend; Africa followed the global trend also, to a certain extent, but with a slightly lower rate in the 'yes, fully' reply (31%) and a slightly higher rate (50%) in the 'yes, somewhat' reply. Latin America and the Middle East both stand out when compared to the global average. Latin America's share of replies to 'yes, fully' was much higher (56%), whereas its replies were lower than the global average in both the 'no' categories.

Additionally, a recent report has pointed out that the temporary cessation of on-site activities of higher education institutions worldwide has greatly interrupted their functioning, and has had a varied impact, since this has depended on two factors: first, the capacity of institutions to stay active in their academic activities, and second, financial sustainability [68,69]. These institutions have made enormous efforts to continue teaching classes online, considering the inexperience in such situations. While the confinement of the population has shown that many activities can be carried out online, technology transfer has not been easy, especially because shortages of digital skills persist in some socioeconomic groups and there are disparities in relation to the access and use of technology. For example, less than half of Latin Americans have experience with computer use and digital skills for basic professional tasks. For this reason, more than half of the population of the region were excluded from remote activities [68].

The reality of Latin America presents great challenges to the governments of the region because these are called to include the technological development of each country in their national policies in accordance with the requirements of globalization, which shows the existing tension between the characteristics of LAC and its ability to respond to an adverse context. Providing a general overview of current AI and the progress achieved by Industry 4.0 in each country in terms of the use of AI, the main findings of the report regarding the higher education were the following:

- All considered countries have a digital strategy. Mexico and Argentina include the ecosystem in the development of a national strategy proposal;
- The lack of digital infrastructure in the region is a key challenge in terms of AI benefits. Regardless of the differences recorded for rural and urban areas, a lack of connectivity is predominant in all the 12 countries studied;
- In terms of gender, schooling, and English proficiency, one woman for every two men in the region participates in a science, technology, engineering, and mathematics (STEM) program. The average level in English is only 56 percent;
- Close to 75 percent of LAC major universities participate in research and development in relation to autonomous systems. Moreover, public and private research centers generate 50 percent of this type of research. Over 96 percent of the main universities in these countries offer degrees related to AI and 50 percent have their own specialized laboratory or center.

### 4.2. Digital Transformation and AI to Support Education: The Case of Chile

In this context, the case of Chile is particularly relevant, which is supported by the following arguments [64]: (1) Chile is the first Latin American country in the Human

Development Indicator, according to the United Nations; (2) According to the International Students Assessment (PISA) exam, Chile is placed in the top position of the region; (3) Chilean Universities are among the most consolidated Higher Education Institutions in Latin America. Digital transformation in Chile started a few years ago and has become a reality in different social areas. However, the COVID-19 pandemic is responsible for causing an acceleration of the process of digitalization of organizations since it forced them to incorporate digital technology to face the new challenges that appeared as a consequence of social distancing and sanitary measures implemented by the government. This crisis period has meant constant learning for Chile since technologies have become increasingly necessary to participate in the new national and international context.

In Chile, digital transformation is measured through the Digital Transformation Index (DTI), which showed that in 2019 Chilean companies were in the digital intermediate category. This is category number three out of five total categories (1. Analogical, 2. Digital beginner, 3. Digital intermediate, 4. Digital advanced, 5. Digital leader). It is essential to highlight that in 2018 Chilean companies were categorized as Digital beginners and moved into the next category in 2019. Between 2018 and 2019 the digital gap decreased by six percentage points [70]. Eight months after the COVID-19 pandemic began, the level of digitization in Chile continued to be at the 'Digital Intermediate' level. However, the index decreased by seven points compared to the previous year, showing progress since 2018 [71] (Figure 10).

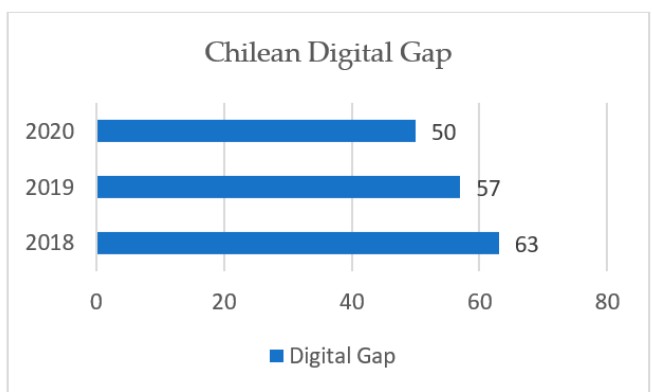

**Figure 10.** Chilean Digital Gap 2018–2019–2020. Source: [69,70] and own elaboration.

This evolution implied that many companies began to generate value through the improvements of their proposals and/or increase their processes' operational efficiency through the use of technology. The category that Chilean companies were at the beginning of the crisis caused by COVID-19 is relevant because the use and application of technology enabled the productive sector to adapt to the contingency and assume its costs [70].

In Chile, the pandemic meant that between March and June 2020, sales fell by an average of 18%, with SMEs being the most affected, registering falls of 25% compared to sales of big companies, which only fell by 16%. The most affected sectors were as follows: Arts and Entertainment with a fall of 69%, Hotels and Restaurants, and Construction, decreasing by 46% and 39% respectively. In contrast, sectors such as Mining, Information and Communications, Electricity and Gas, and Health maintained their sales at a relatively stable rate, with a maximum drop of 2% between March and June 2020 [72]. However, SMEs and enterprises quickly adapted to process automation and teleworking. This ability to adapt remotely was a consequence of the effects of the pandemic and the need for companies to remain operational during the health crisis [69].

The state, for its part, has developed a strategy that allows them to become an actual digital state to respond to the needs of people and companies. To achieve this goal, the Digital Government division was created, whose mission is the digital transformation of the Chilean State. This division depends directly on the Ministry of the General Secretariat of the Presidency. Among the examples of the Chilean agenda are the countries that are

part of the OECD, which are measured through the Digital Government Index, which positions Chile in fourth place compared to the Latin American countries that participated in the study in 2019 [69] (see Figure 11).

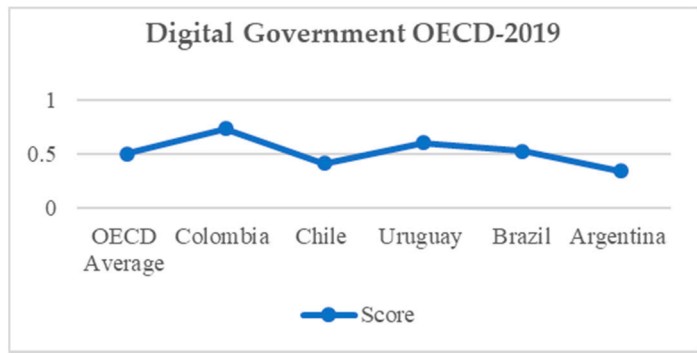

**Figure 11.** OECD Digital Government Average 2019. Source: [69,72] and own elaboration.

The 2019 OECD report has recognized Chile's ongoing commitment to digitization at the state level which has been demonstrated in the authorities' ability to use technology during the pandemic. A concrete example of this use is the virtual police registry that allows citizens to request travel permits in places that are in mandatory quarantine. Another example is the "Mobility Pass" which is a document that is issued virtually to all persons who have completed their COVID-19 vaccination process and who have completed 14 days since the second dose of the vaccine. These changes show the progress that the country has made in the last decade regarding the use of technology [73]. It is expected that with the implementation of the Digital Transformation Law [74], progress will be made in the digitization of interactions between Chileans and the government.

The government of Chile states that the Plan Paso a Paso (Step by Step Plan) is a progressive strategy to face the pandemic, taking into account the health situation of each area in particular. It has five scenarios or progressive steps, ranging from Quarantine to Advanced Opening, with specific restrictions and obligations. The progress or regression from one step to another is subject to epidemiological indicators, the healthcare network, and traceability. As of July 2021, the entire country has advanced in the step-by-step plan, lifting quarantines nationwide (this information is available on https://www.gob.cl/coronavirus/; accessed on 3 August 2021).

In education, the COVID-19 crisis and the sanitary measures implemented by the Plan Paso a Paso (Step by Step Plan) of the Chilean government forced all educational institutions of all levels to stop their on-site activities. For this reason, schools, colleges, and universities decided to continue the academic year online. This situation brought several complications for this sector, especially as each educational institution demonstrated its ability to innovate, adapt, and support teachers and students in different ways. Even though Chile is one of the LAC countries that has showed further technology growth in the region (fourth at the regional level), this has been unequal as its availability is closely linked to socioeconomic levels. In Chile, the less frequent and varied use of the Internet is associated with a low educational level, older age, low socioeconomic level, and being a woman [67]. The best example of the digital divide in this country is shown in the use of the Internet, where the most disadvantaged social groups have fallen behind. This situation had to be faced at that moment. Each university created ways of making their classes accessible to the students. Many universities decided to give their students prepaid chip cards to connect to classes in their homes. This strategy was positive in some cases. Students living in areas where there was no Internet connection were left behind and excluded from online classes. In this way, the COVID-19 crisis revealed the broad social and technology divides in our country. For this reason, the return to face-to-face classes has been part of the public debate throughout the health crisis, mainly because of the persisting gap in the country. The consequences of this gap can clearly be seen in the results obtained

by high school students in the tests carried out by the Education Quality Agency. In these tests, students did not reach 60% of the minimum learning requirements for the year 2020, which can be seen in data obtained from the Comprehensive Learning Diagnosis (EIS). A recent study revealed that 68% of parents believe that their children have learned less with online education. Therefore, developing digital skills and implementing relevant methodologies for online education is pivotal for Chilean education [73].

In short, in the Chilean case, the health crisis increased e-learning, streaming, online shopping, marketing, and teleworking, thus boosting digital transformation and the ability to renew productivity and inclusion in the country, but it also created series of barriers, gaps, and transition costs [73]. An example of this is that Chile is positioned in third place in America in the Competitiveness Ranking carried out by the International Institute for Management Development. It shows that the country lacks a technological infrastructure that allows universal digital transformation, since, currently, only large companies have begun to adopt digital technologies, and not micro-businesses and SMEs, who are far behind (IMD, 2020). The same has happened with people since an increased access to technology is found in groups with greater purchasing power. Thus, to reap the benefits of digitization for all, the state must invest in both infrastructure and education.

Finally, the case of Chile is particularly interesting because in recent years it has made considerable progress towards digital transformation. The state and companies have incorporated technology to become more efficient and SMEs have tried to adapt to the challenges that the pandemic has imposed on them. However, Chile still has a long way to go in order to position itself as a digitized country.

## 5. Conclusions and Theoretical and Managerial Contributions

The increased global presence of technological advancement highlights a rapid transformation in higher education. This type of phenomenon has grown rapidly on a grand scale by leveraging the growth of virtual learning and its global impact in particular. At the beginning of the XXI century, economic power shifted towards digital industries. Higher education institutions' increasing role as major global actors has attracted global attention in both developed and underdeveloped countries. Given recent digital developments, some institutions may choose to temporally refocus on growth in their domestic and international segments and markets. For this reason, this paper aimed to describe these critical and relevant issues. The main conclusions, contributions, limitations, and further research are presented as follows.

### 5.1. Conclusions

This paper illustrated the implementation of digital transformation methodologies among several countries regarding new contexts in higher education. Moreover, we aimed to shed light on this topic by analyzing it at regional level and in Chile in particular, while most of extant research has focused on this consideration by describing it at university level. To conduct our investigation, we exploited a broad data source that was made up of an original survey of university managers, combined with data from the Chilean case. The results of our empirical analysis showed that universities attach more importance to the institutional perspective and the adoption of a particular development strategy. For this reason, these pages focused on describing the evolution and implementation of digitalization processes regarding an important country in South America. In particular, in this paper we have described some variables and indicators which have a relevant function in an educational ecosystem, such as the Chilean case. Moreover, according to the leadership consultation and the comprehensive consultation, the proposition hypothesized in this paper was tested and accepted, and our results and insights can be evidenced in these contexts.

In this research, an extensive landscape was described to understand how the digital transformation operates. This panoramic view suggests that this process has a wide and relevant range of impacts on technological change in higher education institutions. The

analysis also suggested that the vision plays a role in every aspect of the university's organizational structure. The data suggested that the institution's reliance upon earned income allows the university to be selective about the money that it receives in the form of corporate sponsorship, fees, or investments. The Chilean educational ecosystem was used as a framework to better understand how a particular educational environment operates.

In this paper, a relevant finding was that, while all higher education institutions are governed by their respective states and national policies, access to information goes beyond national boundaries; therefore, it is important to have a global platform to exchange and to discuss how new technologies are transforming education. For managers of the educational institutions to understand how a university operates utilizing a technological structure, certain established norms and regulations about the educational ecosystem and field need to be addressed. The first established norm and regulation to be addressed is that higher education institutions are not vision-focused and are only concerned about making utilities. The literature and the data suggested that to be successful, both public and private higher education institutions need to be vision-oriented and intentional about how the staff realize that mission and its vision. Public universities have published missions that are similar in scope to those for private institutions and, as the data from the Chilean case suggested, are used for making both strategic and operational decisions.

Nowadays, higher education environments are experimenting with changes due to an increasing unpredictability. Educational institutions are diverse in nature and scope, and they operate in very different contexts. However, technology has an impact on the skills and competences required by students to take part in society, and on how to access information and knowledge, particularly. Although this transformation is taking place in different ways and by different means and opportunities, one important aspect in this line is how to shape higher education in a digital world.

Not only small- and medium-sized universities, but also other educational models in higher education possess the advantages of flexibility and adaptability, which helps them cope with technological advancements and combine student-focused approaches with the social aims of public and private institutions to maintain competitiveness and sustainability. According to the results here described, the consultation affirmed that a great majority of the respondents in the leadership consultation considered digital transformation a high priority in general.

The open consultation carried out by IAU demonstrated existing inequalities in terms of access for exploring the potential of technology in higher education, for example, in case of internet infrastructure. This constitutes a significant threat to future societies and illustrates the divergence between those who have access and those who have not. The existence of different educational ecosystems has stimulated a debate about the different technological structures available to higher education institutions in Latin America. Essentially, the approach exposed in this study operates using extensive descriptions as well as analysis. This kind of discussion capitalizes on some insights suggesting that some particularities of educational systems appear to be relevant to each institution's mission and vision.

*5.2. Theoretical and Managerial Contributions*

In this paper, the main contributions, related to conceptual, contextual, and managerial implications, were oriented to describe the role of strategic processes led by managers and implemented through a broad range of voices and diverse activities in higher education institutions. Moreover, this paper investigated the actions adopted by universities in performing digitalization activities. Specifically, two indicators, such as the use and application of technology have enabled the educational sector to adapt to the pandemic and assume the costs of it. For this reason, this study was motivated by the consideration that universities' strategic processes in adopting digital transformation should be aligned with more general goals and the missions of the academic institution, and they should reflect its distinctive characteristics. To help universities in their choice, it is fundamental to

identify the range of voices that should guide them. Latin American and Caribbean Higher Education Institutions have much to offer on both historical and environmental heritage from a digital point of view. Provided with digital tools, the challenge is being taken up and the transformation is becoming possible.

This study provides interesting as well as relevant insights about the relationship between university characteristics and their strategic processes, and it does so at the Latin American level. In addition, it contributes to the theoretical and empirical literature on higher education institutions in several ways. On the one hand, we took a step forward in the application of the notion of strategy to universities, thus contributing to the growing literature analyzing this topic. Research in this domain has shown the usefulness of adopting a strategic perspective to accelerate digital transformation over resource acquisition. On the other hand, we contributed to a growing literature emphasizing the role of context in stimulating the extent and variety of activities developed by higher education institutions and their main leaders and managers. That is, while several studies in this domain have looked at the link between institutional determinants and outcome dimensions, we investigated the role of the digital transformation strategy as an important component in the process through a public, open consultation configurating a particular qualitative methodology in nature. By doing so, we documented a broad set of institutional voices on the digitalization strategic choices of higher education institutions. The results and findings of this paper provide an impulse for future research in this field.

Another contribution regarding the educational field is that the use of digitalization and technological systems and activities in higher education leads to vision drift. The data suggested that institutions should adopt a systemic perspective or include certain best practices of digital transformation from other contexts. Additionally, the data from the Chilean case shows that higher education institutions can implement several experiences to manage the digital transformation and allow institutions to receive positive impacts on reputation.

To understand this framework, the literature struggled with some previously held ideas about what certain elements or variables of a particular educational environment model should look like. For example, throughout the process of creating a particular ecosystem, assumptions about operations are challenged and the university can understand what it needs to deliver, create, and provide value. Regarding the two specific objectives related in this paper, our insights reinforce how important it is to evaluate some variables and indicators to manage different cultural, social, environmental, and economic considerations well at higher education institutions. Throughout this paper, digital transformation in Higher Education Institutions has been characterized as a set of dimensions by collecting several sources on an educational digital transformation systemic perspective, and an assessment of the main dimensions for understanding this digital transformation from a systemic perspective in Latin America, in empirical terms, has been offered.

To sum up, this study may be considered as a conceptual and contextual framework of knowledge with application in diverse educational contexts, considering the specific aspects of a Latin American country in particular. This paper includes different experiences captured by agents and institutions. Furthermore, these results contribute to a reinforcement of the relevance of diverse aspects regarding the three main analyses in these type of institutions, that is, the context of digital transformation in higher education institutions, the relevance of innovation in higher education in terms of a systemic perspective, and the necessity to measure the digital transformation of higher education in Latin America in particular. The specific key performance aspects here described reinforce the importance of managing certain instruments in the most important social, cultural, economic, and environmental dimensions. Additionally, some aspects considered as limitations are as follows: for example, the difficulty to coordinate agents and institutions from different geographical locations and the different visions of the institutions involved in the digital transformation project.

In conclusion, the presence of technological projects in different regions and countries facilitates the implementation of a standard of quality in the management of the digital transformation in events and institutions. The repository of knowledge generated by this paper was oriented to show these results and findings to encourage these insights in the future [74–77]. From a comparative perspective, the application of a systemic perspective in this educational context demonstrates the usefulness of focusing on diverse indicators at different levels of management in higher education [78–81].

*5.3. Limitations and Further Research*

Higher education institutions and universities should consider which digitalization model allows the institution to best implement their vision. A university should not adopt a particular structure just because it believes it will make the institution more technologically stable. The analysis and results of this paper pave the way for future research in this field. At the empirical level, the cross-section nature of available data in this paper cannot rule out causality concerns on the relationship between university characteristics and their strategies. However, the opposite might as well happen, thus suggesting that the characteristics of higher education institutions change as a response to a given digitalization strategy adopted.

Universities in Latin America can engage in similar digital activities and can make a relevant social contribution. To sum up, the decision of which technological structure to incorporate should be considered based upon what makes the most sense for the institution.

In terms of the limitations of this research, this is a descriptive study looking at one perspective focused on Chile in particular [81,82]. For this reason, it may not be generalizable to how other countries operate or how other institutions operate outside of the South American region [9,82,83]. Additionally, there were limitations that affected the way data were obtained.

**Author Contributions:** All authors listed have made a substantial, direct, and intellectual contribution to this research. L.M.C.S. and K.N.V. designed the study, wrote the paper, and analyzed the data. K.N.V. and S.Q.y.A. assisted by collecting data. S.Q.y.A. reviewed the entire document. All authors have read and agreed to the published version of the manuscript.

**Funding:** A previous version of this research received the ESIC Best Paper Award in Digital Transformation, 35th Annual Meeting of the European Academy of Management and Business Economics (AEDEM). The APC was partially funded by the Universidad de Las Américas.

**Institutional Review Board Statement:** Ethical review and approval was not required for the study on human participants in accordance with the national legislation and institutional requirements.

**Informed Consent Statement:** Written informed consent from the participant was not required to participate in this study in accordance with the national legislation and the institutional requirements.

**Data Availability Statement:** The datasets generated for this study are available on request to the corresponding author.

**Acknowledgments:** The authors would like to express their gratitude to Universidad de Las Américas, ESIC, and AEDEM for providing technical and financial support in this paper.

**Conflicts of Interest:** The authors declare no conflict of interest.

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
