# Peer review of "An Institutional Perspective for Evaluating Digital Transformation in Higher Education: Insights from the Chilean Case"

_sustainability, doi:10.3390/su13179850_

Round 1

Reviewer 1 Report

The paper is very interesting, especially when the educational problems of Chile are raised. However, the paper is very chaotic. You try to address too many problems in one paper. Frankly speaking, I am lost at the end. Please try to:

  1. In the introduction section to formulate a clear goal of the paper, one or two hypotheses, describe the methods.
  2. Focus your attention on the narrower problem.
  3. Please try to be more specific.

For example, in the "Conclusions" section you state that "In this paper, the main contributions, related to conceptual, contextual, and managerial implications, are oriented to describe some strategic processes leaded by managers and implemented through a broad range of diverse activities in higher education institutions. Some indicators, such as the use and application of technology enabled the educational sector to adapt to the pandemic and assume the costs of it..." There is too many statements like that: some, some...

I am looking at the sentences like that and there are very general, and I cannot find anything substantial. 

For example, you are stating "A university should not adopt a particular structure just because it believes it will make the institution more technologically stable". Why they shouldn't. Do you have any empirical argument to support this statement? Probably the statement is correct, but you need to be cautious about statements like and support them with empirical arguments. 

In line 748 you state "This manuscript includes different experiences captured by agents and institutions." - is it a manuscript or a paper? It looks like taken and cut off from the manuscript, but in a very chaotic way. 

Anyway, I cannot find what methodology you are using? Do you employ any statistical analysis? 

I propose that you select one or two very specific problems and address them in a more thorough and analytical way. 

Author Response

Modifications Letter

Dear colleagues,     

First of all, we would like to thank the reviewers, who made important suggestions to improve the quality of our paper.

We chose to respond to the reviewers’ requests, grouping the questions by reviewer. Our answers are highlighted in bold. In addition, our changes in the revised version of the article are presented in red color mode as requested by the associate editor.

We made a special effort to answer to all the questions raised by the reviewers. Hoping to have answered the suggestions outlined satisfactorily, we are at your disposal to answer any questions, as well as to carry out further revisions as necessary.

Sincerely,

The authors.

Reviewer 1:

  • In the introduction section to formulate a clear goal of the paper, one or two hypotheses, describe the methods. Focus your attention on the narrower problem. Please try to be more specific.

Many thanks for your comments. The new version of the paper has been modified according to your suggestion. We reformulated several phrases, emphasizing the goal of the paper and describing the method in the introduction section. We added clarifications making this section more specific.

2) For example, in the "Conclusions" section you state that "In this paper, the main contributions, related to conceptual, contextual, and managerial implications, are oriented to describe some strategic processes leaded by managers and implemented through a broad range of diverse activities in higher education institutions. Some indicators, such as the use and application of technology enabled the educational sector to adapt to the pandemic and assume the costs of it..." There is too many statements like that: some, some...

Many thanks for your comments. We added and elaborated the conclusions section in order to explain better the implications, contributions, limitations, and further research of this study.

3) I am looking at the sentences like that and there are very general, and I cannot find anything substantial.

Many thanks for your comments. That is another good point, and we totally agree with it. We added numerous elaborations in previous sections, and we included in the last section a detailed review with conclusions, both theoretical and empirical contributions, and limitations and further research. Please, see the new version of the paper.

4) For example, you are stating "A university should not adopt a particular structure just because it believes it will make the institution more technologically stable". Why they shouldn't. Do you have any empirical argument to support this statement? Probably the statement is correct, but you need to be cautious about statements like and support them with empirical arguments.

We agree with the reviewer, many thanks for your comments. The new version of the manuscript has been modified according to them.

5) In line 748 you state "This manuscript includes different experiences captured by agents and institutions." - is it a manuscript or a paper? It looks like taken and cut off from the manuscript, but in a very chaotic way.

We agree with the reviewer, many thanks for your comments. The new version of the manuscript has been modified according to them.

6) Anyway, I cannot find what methodology you are using? Do you employ any statistical analysis?

Many thanks for your comments. The new version of the manuscript has been modified according to them. Materials and methods section has been rewritten and we add some sentences related to your suggestion. Additionally, we included in the introduction section a consideration regarding the qualitative perspective in nature of this paper, in terms of a public, open consultation carried out in the scope of the study.

7) I propose that you select one or two very specific problems and address them in a more thorough and analytical way.

Many thanks for your comments. The new version of the manuscript has been modified according to them.

Reviewer 2 Report

Presenting the digital transformation landscape in Chile is interesting and the general topic of evaluating digital transformation is very important nowadays. However, there are some major issues to be addressed before this paper could be published:

  1. From the title, we expect the paper is concerned with Evaluating Digital Transformation in Higher Education in Chile. While I could find some data about this and also the explanation regarding the whole ecosystem, the paper is not very well organized with this respect - we see analyses before and post-COVID-19, a lot of elements that are international mixed with the ones relevant only to Chile, Higher Education mixed with some institutional perspectives not necessarily directly linked to Higher education, and so on.  At the beginning of the Conclusions section, I could read the authors only claimed to describe some strategic processes. However, a more organized approach would be needed in order for the reader to better understand why all those ideas were taken into account.
  2. Line 336 - it is written that an instrument was created and validated to measure the elements of the research proposal. This could be valuable for the paper and unfortunately, I could not find it in the next lines. How was this instrument validated?!? 
  3. Presenting the study of IAU, starting with Table 1 lines 421-422 is useful in understanding the position of Chile compared within the international context. However, please develop this comparison. Also, the analysis using only percentages of particular responses is not very useful, maybe another method should be used.
  4. Figure 1 - Chilean digital gap - percentages should be placed. Also, is there any chance to see what happened in 2020? (since the digital transformation was for sure higher due to the changes imposed by COVID-19)
  5. There were some sentences that could be found in the study Assessing Digital Transformation in Universities, https://www.mdpi.com/1999-5903/13/2/52/htm. This study was not cited, but only some common sources. Please check and make sure there is not any problem related to this.
  6. Style when citing previous work - "according to [23]" (please write only [23] at the end, without according to, since the names anyway do not appear in the main text; this appeared in many cases - some other examples: supported by [24], proposed by [29], developed by [30], from [31], developed by [32], [36] attempted, [37] developed, [38] emphasized, According to [39], etc. (many others).

Good luck with your work.

Author Response

Modifications Letter

Dear colleagues,     

First of all, we would like to thank the reviewers, who made important suggestions to improve the quality of our paper.

We chose to respond to the reviewers’ requests, grouping the questions by reviewer. Our answers are highlighted in bold. In addition, our changes in the revised version of the article are presented in red color mode as requested by the associate editor.

We made a special effort to answer to all the questions raised by the reviewers. Hoping to have answered the suggestions outlined satisfactorily, we are at your disposal to answer any questions, as well as to carry out further revisions as necessary.

Sincerely,

The authors.

Reviewer 2:

  • While I could find some data about this and also the explanation regarding the whole ecosystem, the paper is not very well organized with this respect - we see analyses before and post-COVID-19, a lot of elements that are international mixed with the ones relevant only to Chile, Higher Education mixed with some institutional perspectives not necessarily directly linked to Higher education, and so on.

Many thanks for your comments. The new version of the manuscript has been modified according to them. We added and elaborated the materials and methods section in order to explain better the methodological process of the study.

  • At the beginning of the Conclusions section, I could read the authors only claimed to describe some strategic processes. However, a more organized approach would be needed in order for the reader to better understand why all those ideas were taken into account.

Many thanks for your comments. The new version of the manuscript has been modified according to them.

  • Line 336 - it is written that an instrument was created and validated to measure the elements of the research proposal. This could be valuable for the paper and unfortunately, I could not find it in the next lines. How was this instrument validated?!?

Many thanks for your comments. The new version of the manuscript has been modified according to them. Given the exploratory and descriptive nature of this research, we added and elaborated the measurements subsection in order to explain better the methodological process of the study.

  • Presenting the study of IAU, starting with Table 1 lines 421-422 is useful in understanding the position of Chile compared within the international context. However, please develop this comparison. Also, the analysis using only percentages of particular responses is not very useful, maybe another method should be used.

Many thanks for your comments. The new version of the manuscript has been modified according to them.

  • Figure 1 - Chilean digital gap - percentages should be placed. Also, is there any chance to see what happened in 2020? (since the digital transformation was for sure higher due to the changes imposed by COVID-19)

Many thanks for your suggestion. In the new version, Figure 1 - Chilean digital gap - the percentages have been placed. In addition, a paragraph has been added referring to the digital transformation in Chile in the 2020 period (covid-19)

  • There were some sentences that could be found in the study Assessing Digital Transformation in Universities, https://www.mdpi.com/1999-5903/13/2/52/htm. This study was not cited, but only some common sources. Please check and make sure there is not any problem related to this.

Many thanks for your suggestion. We present a new version of numeration with common sources and references of empirical studies. Additionally, we included the following reference:

Rodríguez-Abitia, G.; Bribiesca-Correa, G. Assessing Digital Transformation in Universities. Future Internet 2021, 13, 52. https://doi.org/10.3390/fi13020052.

CORFO, Cámara de Comercio de Santiago y PMG Business Improvemen (2020) Índice de Transformación digital de empresas. Tercera versión estudio 2020. Comité de Transformación Digital: Santiago.

  • Style when citing previous work - "according to [23]" (please write only [23] at the end, without according to, since the names anyway do not appear in the main text; this appeared in many cases - some other examples: supported by [24], proposed by [29], developed by [30], from [31], developed by [32], [36] attempted, [37] developed, [38] emphasized, According to [39], etc. (many others).

We revised the manuscript and adapted the method references citation, as recommended by the reviewer.

Round 2

Reviewer 1 Report

In my opinion, the Author / Authors addressed most of my points. Therefore, I am willing to accept the paper in a present form. 

Author Response

Many thanks for your comments.

Reviewer 2 Report

Dear Authors,

I could see an improved version of the paper, congratulations on your work. There are still some minor details that could improve the paper, and the part reflecting some different perceptions in LAC compared to Europe would be very interesting if using stronger methods.

  • With respect to graphs, please write only the idea in the title and not the whole question as it appeared in the questionnaire.
  • Referring to the same part of the paper (perceptions of the developments related to digital transformation), if there is another way in which to use any statistical methods for interpreting respondents' views, please use them (i.e. for comparison between groups).

Good luck with your work.

Author Response

1. There are still some minor details that could improve the paper, and the part reflecting some different perceptions in LAC compared to Europe would be very interesting if using stronger methods.

Many thanks for your comments. The new version of the manuscript has been modified according to them. We added and elaborated the materials and methods, and the results section in order to explain better the methodological process of the study and our specific data analysis.

2. With respect to graphs, please write only the idea in the title and not the whole question as it appeared in the questionnaire.

Many thanks for your comments. The new version of the manuscript has been modified according to them.

 3. Referring to the same part of the paper (perceptions of the developments related to digital transformation), if there is another way in which to use any statistical methods for interpreting respondents' views, please use them (i.e. for comparison between groups).

Many thanks for your comments. The new version of the manuscript has been modified according to them.
